# Fungal Biodeterioration of a Historical Manuscript Dating Back to the 14th Century: An Insight into Various Fungal Strains and Their Enzymatic Activities

**DOI:** 10.3390/life12111821

**Published:** 2022-11-08

**Authors:** Gomaa Abdel-Maksoud, Mahmoud Abdel-Nasser, Mahmoud H. Sultan, Ahmed M. Eid, Saad H. Alotaibi, Saad El-Din Hassan, Amr Fouda

**Affiliations:** 1Conservation Department, Faculty of Archaeology, Cairo University, Giza 12613, Egypt; 2Department of Manuscripts Conservation, Al-Azhar Al-Sharif Library, Cairo 11511, Egypt; 3Department of Botany and Microbiology, Faculty of Science, Al-Azhar University, Nasr City, Cairo 11884, Egypt; 4Department of Chemistry, Turabah University College, Turabah, Taif University, Taif 21944, Saudi Arabia

**Keywords:** historical manuscript, fungal deterioration, hydrolytic enzymes, leather conservation

## Abstract

This study aims to assess the deterioration aspects of a historical manuscript dating back to the 14th century that was deposited in the Library of the Arabic Language Academy, Cairo, Egypt. The study aims at the exploration of the role of various fungal strains that had colonized this deteriorated manuscript in its biodeterioration through their efficacy in the secretion of various hydrolytic enzymes. To evaluate the deterioration, various techniques, including visual inspection, attenuated total reflectance Fourier transform infrared (ATR-FTIR), scanning electron microscopy (SEM), X-Ray diffraction analysis (XRD), color change, and pH value, were utilized. The fungal strains linked to the historical document were isolated, identified, and evaluated for their deterioration activities. The findings demonstrate that the manuscript exhibits a variety of deterioration signs including color change, brittleness and weakness, erosion, and removal of the grain surface pattern in leather binding. According to the ATR-FTIR, the chemical composition of the historical paper and leather underwent some alterations. The historical paper has a lower level of cellulose crystallinity than the control sample. *Penicillium chrysogenum* (two isolates), *P. citrinum* (four isolates), *Aspergillus ustus* (three isolates), *A. terreus* (two isolates), *A. chinensis* (one isolate), *Paecilomyces* sp. (one isolate), and *Induratia* sp. (one isolate) were among the fourteen fungal strains identified as being associated with the historical manuscript. These fungal strains produced several hydrolytic enzymes with high activity, such as cellulase, amylase, gelatinase, and pectinase, which play a key role in biodegradation.

## 1. Introduction

The preservation of manuscripts is crucial since they are among the most significant sources of information about our culture, science, politics, economy, and history [1]. The degradation of historical paper and leather binding during natural aging is influenced by several factors, including their composition, the process used to make them, the length of time they were exposed to the environment, humidity, temperature, light, pollution, microorganisms, and environmental quality [2,3,4].

The development of paper and leather conservation procedures depends mainly on the research of deterioration mechanisms. The fluctuation in relative humidity affects papers and leathers since they are hygroscopic materials (they can absorb and release water depending on relative humidity and temperature in the surrounding environmental conditions of manuscripts). As a result, various deterioration-related characteristics such as contracting, wrapping, cracks, dryness, and weakening are manifested [5,6,7]. Wilson [5] has reported that photooxidation causes paper to deteriorate. Sulfur dioxide and nitrogen dioxide promote photooxidation, which increases with rising relative humidity. Paper and leather fade in the presence of light. Sulfur dioxide, nitrogen oxide, and ozone are chemicals that can be hazardous to library items [5]. In addition, the author reported that various pollutants have been identified as contributing to the destruction of archeological manuscripts, including formaldehyde, aldehyde, acetic acid, mineral acids, hydrocarbons, and sulfur dioxide from protein-based glues. Acids had a stronger impact as temperature and relative humidity increased. The most typical signs of deterioration caused by pollution on old leather and paper are weakness and occasional discoloration. As a result of colony formation and fungal pigments, many fungal strains can occupy, modify, and deteriorate all kinds of organic and inorganic materials in museums or storage facilities, seriously tarnishing their aesthetic quality [8].

Leather is an organic substance that contains many nutrients for microorganism growth, according to Abdel-Maksoud et al. [9] The chemical and structural nature of the substrate as well as environmental conditions are significant parameters that affect the quantity and quality of microbial colonization on works of art. Additionally, they claimed that *Aspergillus niger*, *A. flavus*, *A. terreus*, *A. ochraceous*, *A. carbonarius*, *A. fischeri*, *A. fumigatus*, *Penicillium notatum*, *P. oxalicum*, *P. rubrum*, and *Aletrnaria alternata* were able to be isolated from historical leather. According to Sequeira et al. [1], paper is thought to be highly susceptible to fungi. This is because the paper is hygroscopic and contains plentiful carbon sources for these heterotrophic organisms (cellulose, hemicelluloses, lignin, adhesives, and sizing’s). Numerous species of fungi, including *A. niger*, *A. terreus*, *A. ustus*, *Alternaria* sp. *Penecillium* sp., and others have been isolated from historical papers.

Filamentous fungi have an important role in the production of proteases, which are capable of hydrolyzing a wide variety of proteins in addition to collagen [10]. Additionally, they produce several cellulases that hydrolyze the main components of the paper and degrade it [11]. Aside from that, enzymes including gelatinases, xylanases, and amylases play a crucial part in the biodegradation of paper’s contents [12]. Extracellularly produced microbial enzymes such as amylases, pectinases, lipases, xylanases, and laccases, as well as hyphae from fungal development, can exert mechanical stress on the paper and weaken it [13,14]. It must be made clear that air pollutants or acids produced by fungi are significant contributors to the historical manuscript’s destruction [15]

One of the most crucial factors for conservators when determining the conditions for preserving paper manuscripts and binding leather books in libraries and archives is the study of fungi and their efficiency in producing enzymes that degrade the components of paper and historical leather [16,17].

The current study aims to assess the deterioration aspects of a historical manuscript (paper and leather bookbinding) that dates back to 1327 AD and is kept in the Library of the Arabic Language Academy in Cairo, Egypt. The study also aims to investigate the role of a fungal community isolated from this historical manuscript in its biodeterioration. To accomplish this, a variety of study techniques were applied, including visual inspection, SEM, ATR-FTIR, XRD, color changes, and pH values, to look at various elements of deterioration in the chosen historical document. By using conventional and molecular techniques, many fungi connected with the deterioration of old paper and leather bookbinding have also been isolated and identified. To further understand their role in biodeterioration, the ability of these fungal strains to produce many extracellular hydrolytic enzymes, such as cellulase, amylase, pectinase, and gelatinase, were investigated.

## 2. Materials and Methods

### 2.1. Materials

#### 2.1.1. New Whatman Paper (Control)

Whatman grade one filter paper was used as a control for all analyses and investigations used in this study. It has a neutral pH value, is free of impurities, and has no additives. The Whatman grade one paper was made of cotton, and the paper sheets of the historical manuscript are also made of cotton.

#### 2.1.2. New Vegetable-Tanned Leather Samples (Control)

Vegetable-tanned (mimosa) goat skin samples were used as a control for leather bookbinding of the historical manuscript and prepared according to Abdel-Maksoud [18]. It was used as a reference (control) because it was used in the bookbinding of the studied manuscript. The historical manuscript’s deteriorating process can be investigated by contrasting the characteristics of the new and old leathers.

#### 2.1.3. Historical Manuscript Studied

The historical manuscript under study is named Musnad of Imam Muhammad bin Idris al-Shafi’I. The manuscript was written in 1327 A.D. and preserved at the Library of the Arabic Language Academy in Cairo, Egypt. The leather binding of the manuscript was covered by marble paper. This manuscript was written in the Arabic language, contains one volume with a size (cm^3^) of 28 × 19 × 5 (length × width × height), and is composed of 293 paper sheets.

RS485 Modbus was used to encode the temperature degree and relative humidity (RH) value (RS-WS-N01-6, Jinan, China). The RH was 65% and the temperature around the document under study was 24 °C. In the absence of an air conditioning system, the manuscript was preserved on iron shelves with iron cases. The storage cabinet was kept in a basement below ground level and next to the Nile; as a result, there is very little ventilation and a high relative humidity level.

### 2.2. Methods

The following methods of investigation and analysis were used. It should be mentioned that the samples were always taken from damaged places with severe aspects of deterioration. This will give a clear indication of the actual damage found in the studied manuscript.

#### 2.2.1. Photographic Documentation

The deterioration detected on the surface of the historical paper and leather binding of the historical text was photographed with a digital camera (Samsung camera 38MP, f/2.2 lens slot) and described with the naked eye.

#### 2.2.2. ATR-FTIR Analysis

ATR-FTIR analysis was used to study the chemical change in functional groups present in the historical paper and leather binding. The FTIR spectra of two samples for new and historical paper (the sample was taken from the upper part of the fungal-stained paper) and two samples for new and historical leather (taken from fallen fibers next to the fragile backbone of the bookbinding) were measured using a Bruker Vertex 70-platinum ATR spectrometer with crystal diamonds in the range of 4000–400 cm^−1^ and at a resolution of 4 cm^−1^. The analysis was achieved at the Antiquities Research and Conservation Center—Supreme Council of Antiquities—Ministry of Tourism and Antiquities, Cairo, Egypt.

#### 2.2.3. SEM Analysis

The change in the surface morphology for new and historical samples was analyzed using a Quanta 3D 200i made by FEI-accelerated voltage 20.00 kV and the range of magnification orders was between 250 to 2000×. A minute amount of stained paper (one sample) and leather (one sample) were taken from the fallen samples next to the manuscript studied. The observations were made on the studied samples without any preparation at low vacuum. The investigation was achieved at the Grand Egyptian Museum, Conservation Center (GEM.CC), Giza, Egypt.

#### 2.2.4. XRD Analysis

X-ray diffraction analyses for Whatman paper (control) and historical paper (a minute amount of one sample was taken from brittle stained paper fallen next to the manuscript studied) were performed at the Grand Egyptian Museum—Conservation Center (GEM.CC), Giza, Egypt, using an X-ray diffraction meter system model with the following attributes: Panalytical X, pert pro-PW 3040/60; scan axis: gonio; anode material: Cu; generator settings: 40 mA, 45 kV; and goniometer radius [mm]: 240.00. The crystallinity index was determined by the following equation [19]:(1)$$I_{\mathrm{Crys}.}=\frac{I_{002}-I_{\mathrm{am}}}{I_{002}}\;\times \;100$$
where I_Crys_ = crystallinity index, I_002_ = intensity at approximately 2 θ = 22.6°, and I_am_ = intensity at approximately 2 θ = 19°.

#### 2.2.5. Measurement of Color Change

Color changes of historical paper and leather compared to control samples (the color change was measured on the stained paper, and on the weak lower part of the bookbinding without sampling) were measured using the CIELAB system. The measurement was made using portable Mini-Scan EZ spectrophotometer by Hunter Lab-Reston Virginia (USA). as described by Abdel-Maksoud and Marcinkowska [20]. The CIE system has one channel for the detection of lightness (L*) and two further channels, one for measuring the color change from red to green (a*) and another for measuring the color change from yellow to blue (b*). Overall color difference (DE) was measured according to the following equation [21,22]:(2)$$\Delta E=\sqrt{{\left(\Delta L\right)}^{2}+{\left(\Delta a\right)}^{2}+{\left(\Delta b\right)}^{2}}$$
where ΔL, Δa, and Δb were calculated as the difference between the values of L*, a*, and b* for control and historical samples. This analysis was completed at Cairo University’s Faculty of Archaeology’s Conservation Department.

#### 2.2.6. Measurement of pH Value

##### Measurement of pH Value of the Leather Samples

The pH values of two measurements for new and historical leather bookbinding samples were measured according to Wouters et al. [23], and Abdel Maksoud [24] with some modifications. Mechanically, a sample (0.025 g) of the leather taken from the backbone of the bookbinding was removed from as close to the damaged location as possible in the form of loose fibers. To allow the ions to transfer into the solution, the leather samples were soaked in deionized water for about 6 h.

The pH value was determined using the waterproof pH testers AD11 and AD12. At 21–22 °C, calibration was between 2 and 7. The measurement was undertaken at Cairo University’s Faculty of Archeology, Department of Conservation, in Giza, Egypt.

##### Measurement of the pH of the Paper Samples

The pH value of two measurements for the new and historical paper samples was determined according to Abdel-Maksoud [24] in a non-destructive way without sampling using Adwa AD 1030 pH/mV with flat electrode. The pH measurements were performed by placing the flat electrode on the paper. The measurement was achieved at the Center for Papyrus and Inscription Studies, Ain Shams University, Egypt.

#### 2.2.7. Microbiological Analysis

##### Fungal Isolation

To isolate several fungus species, the sterilized cotton swabs were placed over the severely degraded area of the historical document before being transported immediately to the lab. Each swab was inoculated separately in the Czapeck yeast extract broth (CYB) media (containing g L^−1^: sucrose, 30; NaNO_3_, 2; KH_2_PO_4_, 1; KCl, 0.5; MgSO_4_.7H_2_O, 0.5; FeSO_4_.7H_2_O, 0.002; yeast extract, 5; Distilled H_2_O, 1 L) supplemented with chloramphenicol (500 mg L^–1^) to suppress the bacterial growth. The inoculated broth media was incubated at 27.0 ± 2 °C for 48 h under shaking conditions (150 rpm) to enhance the fungal growth associated with the historical paper before growing in solid media [25]. After that, approximately 50 µL from the previous enrichment broth media was put onto the surface of the Czapeck yeast extract agar (CYA) plate, which was spread evenly with a sterile glass spatula followed by incubation at 27 ± 2 °C. Daily checks were made on the previously inoculated plates to check for fungal growth, which was then re-inoculated onto a new plate for additional purifications. The purified fungal species were ultimately maintained in CYA and stored in a refrigerator for further study [26].

##### Fungal Identification

The purified fungal species were identified by the microscopic and culture characteristics according to standard keys for *Aspergillus* spp. [27], *Penicillium* spp. [28], *Pacilomyces* sp. [29], and *Induratia* sp. [30]. The molecular identification was achieved by the amplification and sequencing of the ITS region. DNA extraction was achieved according to Khalil et al. [31]. The fungal genomic DNA was purified using Gene Jet Plant genomic DNA purification Kit (Thermo) protocol. The amplification of genes was conducted by PCR (polymerase chain reaction) using purified DNA as a template and ITS primers (ITS1 (5′-TCCGTAGGTGAACCTGCGG-3′) and ITS2 (5ʹ-TCCTCCGCTTATTGATATGC-3′)). The PCR mixture (50 μL) contained; 0.5 μM of each primer, Maxima Hot Start PCR Master Mix (Thermo), and 1 μL of extracted genomic DNA. The PCR protocol was achieved using a DNA engine thermal cycler by Sigma Scientific Services Company (Cairo, Egypt) under the following conditions: 94 °C for 3 min, followed by 30 cycles of 94 °C for 30 s, 55 °C for 30 s, and 72 °C for 1 min, followed by a final extension performed at 72 °C for 10 min. The commercial sequencing was conducted using ABI 3730XL DNA sequencer at GATC Company (Germany). The obtained ITS sequence was compared with those deposited in the GenBank database using the NCBI BLAST program, and the phylogenetic tree was conducted by a neighbor-joining method in the MEGA v6.1 software (www.megasoftware.net, accessed on 5 November 2022) with confidence tested of bootstrap analysis (1000 repeats). The obtained sequences were deposited in GenBank under accession number ON527926—ON527932.

#### 2.2.8. Extracellular Enzyme Activities

Each purified fungal strain was inoculated into mineral salt agar media (MSA) (containing g L^–1^: KCl, 5; NaNO_3_, 6; MgSO_4_.7H_2_O, 0.5; KH_2_PO_4_, 1.5; ZnSO_4_, 0.01; FeSO_4_, 0.01; agar, 15, Dis. H_2_O, 1 L) supplemented with a specific substrate. The activities of amylase, cellulase, pectinase, and gelatinase enzymes were estimated by inoculating the fungal strain on the MSA media supplemented with 1% soluble starch, carboxymethyl cellulose, pectin, and gelatin, respectively, and incubated for 96 h at 25 ± 2 °C. The results were recorded as a diameter of a clear zone (mm) which is calculated by subtracting the diameter of fungal growth from the diameter of all clear zones. The results were recorded after flooding the inoculated plates with an iodine solution to examine the activity of amylase, cellulase, and pectinase, whereas the acidic mercuric chloride was used to examine the gelatinase activity [32]. The experiment was performed in triplicate.

#### 2.2.9. Statistical Analysis

All collected data in the current study are presented as the means of three independent replicates and analyzed using ANOVA analysis by a statistical package SPSS v17. The mean difference comparison between the treatments was analyzed by the Tukey HSD test at *p <* 0.05.

## 3. Results and Discussion

### 3.1. Assessment of the Deterioration Aspects of the Selected Historical Manuscript

#### 3.1.1. Photographic Documentation

The digital camera photography (Figure 1A) showed that the paper and leather binding of the manuscript had some signs of deterioration, including darkening, stiffness, loss of elasticity, and missing pieces. These indicators of deterioration could be the result of the diverse composition of leathers changing and deteriorating as well as their sensitivity to environmental conditions (contaminants, light, humidity, temperature, and pollutants), as previously indicated [33,34,35]. The degeneration of leather may eventually cause the material to disintegrate and damage the document. According to Sebestyén et al. [36], both internal and extrinsic causes could be responsible for some aspects of leather degeneration. The internal elements result from the manufacturing process of leather artifacts. While the external factors are brought on by biological factors (microorganisms and insects), environmental elements (light irradiation, relative humidity, temperature, chemical pollution, and particulate matter pollution), and natural occurrences, including those brought on by climate change [37]. Additionally, several stains were obtained, which, as previously confirmed [38,39], may be the result of an attack by microbes.

For the historical paper manuscript (Figure 1B–E), some signs of deterioration have appeared, including yellowing, fragility, and weakness of the paper. Helmi et al. [40] have reported that the deterioration signs in a historical paper may be due to an increase in the acidity caused by an external factor (pollutants from the surrounding environment), or internal factors. In addition, cellulose can produce acids, such as formic, acetic, lactic, and oxalic acids, as a result of unfavorable environmental conditions. Strong intermolecular bonds prevent the paper from easily releasing these acids, but the moisture continuously shortens the glucose chains. The hydrolysis reaction causes the production of more acids, which enhances the degradation process. It was also noted that the paper’s ink had warped and that water had left spots on it. This type of degradation demonstrated how long-term exposure to high relative humidity had damaged the manuscript under study. There is a chance that certain microbes are responsible for the stains on the paper. Microbial growth is induced by all of the aforementioned factors.

#### 3.1.2. ATR-FTIR Analysis

A.ATR/FTIR Analysis of Paper Samples

The results (Figure 2A) demonstrate that characteristic bands of cellulose functional groups can be seen in the regions of valence vibrations 3332 cm^−1^ (for Whatman filter paper) to 3330 cm^−1^ (for historical paper), with low intensity in the historical sample (0.024) than Whatman filter paper (which was made from cotton fibers as shown by SEM) (0.044). This band indicates the presence of hydroxyl groups and is assigned to the stretching OH vibrations. These findings provide evidence for both intra- and intermolecular hydrogen bonding [41]. The bands at 2899 cm^−1^ and 2907 cm^−1^ with intensity 0.022 and 0.0098 for Whatman paper and historical paper samples, respectively, refer to stretching vibrations of methyl and ethyl groups v(CH_3_) and v(CH_2_) that conceder as cellulose compounds [42]. The characteristic peak at 1644 cm^−1^ and 1634 cm^−1^ with intensity 0.013 and 0.025 for Whatman paper and historical papers, respectively, refer to a physically absorbed water molecule. The strong affinity of paper for water makes it difficult to perform an FTIR study on paper samples, according to Cocca et al. [43]. In addition, they noted that the band of the absorbed and bound water is located in the carbonyl group region (between 1635 and 1670 cm^−1^), and that occasionally it can be rather broad and obscure the bands of the carbonyl groups.

The bands at 1578 cm^−1^, 1541 cm^−1^, and 1508 cm^−1^ with intensities at 0.011, 0.010, and 0.009, respectively, appeared in the Whatman paper samples but disappeared in the historical paper samples. This might be because the historical paper manuscript was subjected to harsh conditions over a long period, affecting its chemical stability and causing the loss of some cellulose functional groups [44]. It can also be argued that this might be due to the oxidation process. This finding was confirmed by Cocca and co-authors, who said that the regions between 1500–1800 cm^−1^ indicated the effect of aging on paper that induced various oxidation processes on paper items [43]. Librando et al. [44] have reported that atmospheric pollutants had promoted various deterioration reactions, especially in association with water. Sulfur compounds are considered the most harmful substances to cause oxidation. They also reported that dust increases chemical and physical damage by absorbing water vapor and pollutants, and by attaching microorganisms. A high concentration of total suspended dust in archives and libraries accelerate the aging and disintegration of paper, especially when there is also high relative humidity.

The bands at 1427 cm^−1^ and 1417 cm^−1^ with intensities of 0.025 and 0.056 appeared for the Whatman paper and historical paper samples, respectively. The bands at 1360 cm^−1^, 1334 cm^−1^, and 1279 cm^−1^ with intensities of 0.027, 0.032, and 0.024, respectively, only appeared for the Whatman paper and vanished with the historical paper sample. The bands at 1314 cm^−1^, and 1317 cm^−1^ with intensities of 0.036 and 0.031 appeared for Whatman and historical paper samples, respectively. The bands at 1248 cm^−1^ and 1204 cm^−1^ with intensities of 0.020 and 0.019 appeared for the Whatman paper and historical paper samples, respectively. All of these bands (from 1427 cm^−1^ to 1204 cm^−1^) are distinct from cellulose fibers and refer to C-O-H, CH_2_ bending vibrations [41].

The bands at 1159 cm^−1^ with intensities of 0.044 and 0.030 appeared for both the Whatman paper and historical paper samples, respectively. The bands at 1108 cm^−1^ and 1053 cm^−1^ with respective intensities of 0.067 and 0.107 appeared with the Whatman paper sample while the bands at 1107 cm^−1^ and 1054 cm^−1^ with respective intensities of 0.046 and 0.069 appeared with the historical paper sample. Those bands (from 1159 cm^−1^ to 1053 cm^−1^) are distinct from cellulose fibers and refer to various C-O stretching, C-O-C, and C-C-O bending vibrations [41]. Other characteristic signals related to the cellulose structure were the bands at 897 cm^−1^ and 874 cm^−1^ with intensities of 0.44 for both Whatman and historical paper samples, and which referred to C-O-C, C-C-O, and C-C-H deformation modes and stretching vibrations in which the motions of the C5 and C6 atoms were at 900 cm^−1^. The bands at 662 cm^−1^ and 660 cm^−1^ with respective intensities of 0.077 and 0.054 appeared for both Whatman and historical paper samples and referred to the C-OH out-of-plane bending mode at 666 cm^−1^.

A clear indication of the above-mentioned deterioration in the historical paper is the absence of specific bands in the samples of historical paper and the rising or decreasing of their intensities in comparison to the Whatman paper (control).

B.ATR/FTIR Analysis Leather Samples

The collagen-based materials spectra are characterized by amide A at the band 3294 cm^−1^ with an intensity of 0.042 for the new vegetable-tanned leather, and at the band 3295 cm^−1^ with an intensity of 0.019 for historical binding [45]. Amide B appeared at the bands 2921 cm^−1^ and 2919 cm^−1^ with respective intensities of 0.026 and 0.010 for new vegetable-tanned leather and historical binding (Figure 2B). The bands of amide A and amide B are associated with the stretching of peptide N-H groups involved in inter-chain hydrogen bonding [46].

Amide I appeared at the band 1633 cm^−1^ and 1622 cm^−1^ with respective intensities of 0.090 and 0.035 for new and historical leather binding, respectively. The amide I bands correspond to the C=O stretching vibration of peptide bonds along the polypeptide backbone with a small contribution from N-H in-plane bending [47]. The amide I band in the region of 1633 cm^−1^ was assigned to the preserved collagen, and was attributed to the unfolding of the helices, but at 1622 cm^−1^ for the historical binding was assigned to gelatin [34]. Amide II (at the bands 1546 cm^−1^ and 1542 cm^−1^ with respective intensities of 0.071 and 0.029 for both new and historical leather binding, respectively) is associated with N-H bending and C-N stretching vibration. Amide III (appearing at band 1280 cm^−1^ with an intensity of 0.053 for the new vegetable-tanned leather, but which disappeared for the historical binding) was associated with N-H in-plane bending and CH_2_ wagging vibration of glycine backbone and proline side chain [48]. The band at 1453 cm^−1^ with an intensity of 0.061 for new vegetable-tanned leather refers to the vC=C aromatic ring. The band at 1453 cm^−1^ with an intensity of 0.029 for historical leather binding may refer to calcium carbonate. Vichi et al. [35] have stated that calcium carbonate is commonly found in historical leather resulting from the carbonatation of calcium hydroxide residues of the liming bath with the CO_2_ present in the atmosphere or added during manufacturing. The band at 847 cm^−1^ with an intensity of 0.043 for the new vegetable-tanned leather, and at 875 cm^−1^ with an intensity of 0.038 for the historical leather binding refer to (δ CO_3_). Vichi et al. [35] have argued that the position of CO_3_ of the historical binding could have indicated that other carbonate species were present. The band at 1032 cm^−1^ in the historical binding may refer to the presence of hydrated gypsum (CaSO_4_·2H_2_O). The powdered gypsum was sometimes used in animal skin processing to remove excess fat [35]. The presence of calcium sulfate hydrated in leather has also been linked to the reaction of sulfate ions with residual calcium milk used in the unhairing process of animal skin.

#### 3.1.3. SEM Analysis

The data (Figure 3A) from the control sample show that the surface of goat skin leather was smoothed and glossed. Jawahar et al. [49], who found that goat skin is distinguished by trios configurations of large pores surrounded by small pores in a cluster, confirmed this finding. The historical sample’s results (Figure 3B) indicate that it was similar to goat skin through its distinctive grain surface pattern. However, it was seen that the surface was rough and damaged, which may have been caused by the impact of the surrounding environmental conditions’ pollution and dust. It might also be the result of repeatedly handling the manuscript incorrectly. Some fine lines were also observed, which may be due to the effect of insects. Furthermore, the leather deterioration could be attributed to the colonization of fungi. Orlita [50] has reported that various fungal strains such as *Alternaria alternata*, *Aspergillus oryzae*, *A. niger*, *A. versicolor*, *A. wentii*, *Chaetomium globosum*, *Cladosporium sphaerospermum*, *Cladosporium herbarum*, *Fusarium chlamydosporum*, *Paecilomyces variotii*, *Penicillium commune*, *P. glabrum*, *P. funiculosum*, *P. verrucosum*, *P. ochrochloron*, *P. rubrum*, *Trichoderma viride*, and *Verticillium tenerum* have the efficacy to deteriorate and deform leathers. As shown, the obtained fungal strains in the current study were similar to these strains especially *Aspergillus* spp. and *Penicillium* spp.

The results obtained from SEM (Figure 3C) show the fiber shape and structure of Whatman paper. It is a ribbon-shaped fiber with some twists in the form of a spiral, and these twists sometimes go to the right and sometimes to the left. Wang et al. [51] have also reported that the native cotton fibers showed a twisted ribbon-like shape and had a relatively smooth surface, primarily because of the wax coating on the fiber surface. For the historical paper (Figure 3D), the fibers used were made from cotton, and their features were similar to the fibers obtained from the Whatman paper samples. Some signs of deterioration were observed including erosion of the fibers, dust found on the surface, and mechanical weakness. These signs may be due to the effect of fluctuation in relative humidity and temperature, light, pollutants, and particulates from the surrounding environmental conditions on the manuscripts. Fungal deterioration was also noticed (Figure 3E), since mycelium, hyphae, and spores of fungi were observed. Additionally, some cracks were also found, possibly as a result of the combined effects of the previously mentioned factors. Borrego et al. [52] have indicated that fungi can affect valuable documents aesthetically, chemically, and mechanically by producing pigments, enzymes, and acids. The environment around the manuscript, particularly at RH levels of more than 65%, temperatures higher than 22 °C, and contaminants, etc., can also be a reason to encourage the growth of microorganisms. Since fungi release spores, which are easily spread by moving air, they are seen as being more harmful than bacteria [53].

#### 3.1.4. XRD Analysis

The measurement of cellulose crystallinity is considered one of the most important properties of cellulosic materials. It gives information about their mechanical, physical, and chemical properties [54]. In this study, the Whatman paper sample (control) was selected because the historical sample was made of cotton fiber which makes for a good comparison. In general, paper made of cotton fibers contains greater than 95% cellulose and has a high degree of crystallinity. The obtained results (Figure 4) show that the crystallinity of the control sample (Whatman paper) was 82.35, while the crystallinity of the historical paper manuscript was 71.43.

It has been indicated that the investigated manuscript’s cellulose had a lesser crystallinity than the control sample (Whatman paper). This result may be the result of the influence of several factors, including light, temperature, relative humidity, air pollution, and microorganisms, which impacted the crystalline-sized portions of cellulose and caused its weakness and fragility [55]. The pH values also demonstrate that the historical sample has more acidity than the control sample. The mechanism of manuscript damage is most greatly influenced by the oxidation process. According to Sandy et al. [56], deteriorated cotton fibers have a lower crystallinity than new cotton fibers. They reported that the crystallinity of degraded papers was a result of natural heat aging with low moisture content recorded at 47.5 to 54%, which was lower than the crystallinity of the new cotton sample.

#### 3.1.5. Measurement of Color Change

The data obtained (Figure 5A) show that the lightness of new vegetable-tanned leather was 72.8, while the lightness of the historical leather binding was 48.5. The reduction was 33%, and the color tended to the dark color. The measurement of a* value proved that the color was red. The historical leather was darker than the new leather. The increase in the red color of the historical sample was 56%. The yellow color (b* value) also increased in the historical leather sample compared with the new leather sample. The darkness in color was also observed from the results of the total color difference, which was 35.91. This value indicates the high deterioration of historical leather.

The results obtained for the historical paper (Figure 5B) show that the lightness of the historical sample was lesser than the Whatman paper sample. The a* and b* values of the Whatman paper were less red and yellow, while the red and yellow colors increased in the historical paper sample. The total color difference of the historical paper was 24.85, which reflects the more severe deterioration of the historical paper compared with the Whatman paper.

The changes in the color values and the total color difference might be due to the effect of light. Historical leather and paper materials are considered very sensitive to light. Natural or artificial lights contain ultraviolet radiation (UV), which provides energy to fuel the chemical reactions that lead to deterioration. UV causes the yellowing, weakening, and/or disintegration of materials. The light also contains infrared radiation, which heats the surface of objects and thus becomes a form of incorrect temperature (too high). The effect of heat in combination with the effect of light will lead to chemical damage by oxidation or by photochemical deterioration. It has been reported that light can lead to a decrease in the degree of polymerization of the cellulose chains, accordingly, weakness can be obtained [57].

#### 3.1.6. Measurement of pH Values

The pH value (Figure 6) of the new vegetable-tanned leather sample was 4.1, while the pH of the historical leather sample was 3.0. Lama et al. [58] have reported that acid deterioration of historical leathers might be derived from the leather manufacturing process. The increase in the application of acid may accelerate the acid-hydrolysis and cause rapid deterioration of the historic leather. They also mentioned that the rapid industrialization during the 19th century and the use of coal gas were considered the major contributors to environmental pollutants, including nitrogen dioxide (NO_2_) and sulfur dioxide (SO_2_). These gases are converted to acid in the presence of water when absorbed by leather as shown below [58]:(3)$$\mathrm{Sulfur}\;\mathrm{dioxide}\;(SO_{2})\;\to \;\mathrm{Sulfur}\;\mathrm{trioxide}\;(SO_{3})\;\to \mathrm{Sulfuric}\;\mathrm{acid}\;(H_{2}SO_{4})$$

It can also be noticed that the acid hydrolysis of the collagen molecules may result from the formation of a hydronium ion (H_3_O^+^) in an acidic environment. NO_2_ in the atmosphere can function as free radicals, speeding up oxidation and the subsequent degradation of historic leather.

Lama et al. [59] have indicated the deterioration mechanism of the leather. They indicated that collagen is the main structural protein in leather. The triple helical structure of the collagen is stabilized by hydrogen bonds. While the formation of a hydronium ion (H_3_O^+^) in an acidic environment may cause acid hydrolysis, break down the hydrogen bonds, and weaken the helical structure. Additionally, the oxidative and hydrolytic reactions are influenced by environmental conditions such as water, heat, pH, light, and gasses. The deterioration by acid hydrolysis might be faster than by oxidation.

The pH value of the Whatman paper (Figure 6) was 7.4, but the pH value of the historical paper was 5.4. Various reasons are responsible for the decreasing pH value of the historical papers as follows. The papers are made from pure cellulose which have a high efficacy to generate organic acids due to natural deterioration [60]. These acids lead to the oxidation of the reducing ends of the cellulose molecules. Furthermore, air pollutants including sulfur oxides (SO_x_) and nitrogen oxides (NO_x_) are an important factor in the increasing acidity of historical paper. These gases can directly oxidize cellulose to produce organic acids, or react with moisture that adsorbs on paper fibers and produce nitrous and sulfuric acids. The hydrolytic deterioration of cellulose in paper fibers, as catalyzed by hydrogen ions, is accepted as the most important cause of book paper deterioration in most libraries [61]. The rate of deterioration increases with a decrease in the pH value (increase in acidity) of a water extract of any paper.

Moreover, acids or enzymes are produced by some microorganisms such as fungi and/or bacteria. Several cellulolytic fungi such as *Alternaria* sp., *Aspergillus* sp., *Fusarium* sp., *Humicola grisea*, *Myrothecìlim verrucaria*, *Penicillium* sp., *Stachybotrys* sp., *Stemphylium* sp., *Trichoderma* sp., *Ulocladium* sp. and *Chaetomium* sp. have been isolated from deteriorated historical books/documents and exhibited a critical role in biodeterioration by secretion of acids and enzymes [62].

Also, iron ink plays an important role in the decreasing pH value of historical paper. Marín et al. have explained that acid hydrolysis and iron(II)-catalyzed oxidation of cellulose are two well-studied processes that take place in the deterioration of historical manuscripts as reported previously [63], through direct acid hydrolysis and/or iron(II)-catalyzed oxidation of cellulose.

Overall, the obtained results of the high acidity of historical samples could be the main reason for various deterioration aspects such as embrittlement, darkening, fragility, etc.

### 3.2. Fungal Isolations and Identifications

The fungal species that inhabit historical leather bookbinding and paper were isolated. Herein, fourteen fungal isolates were obtained from the collected deteriorated samples and designated as AM.1–AM.14. The purified fungal isolates were initially identified by traditional methods using the cultural and microscopic examination. Data reveal that all the obtained isolates belonged to the division Ascomycota. As shown, nine fungal isolates (AM.1, AM.2, AM.3, AM.4, AM.5, AM.6, AM.10, AM.11, and AM.14) were isolated from leather bookbinding, whereas the remaining isolates (AM.7, AM.8, AM.9, AM.12, and AM.13) were obtained from collected deteriorated papers. The primary identification showed that AM.1 and AM.11 were identified as *Penicillium chrysogenum*. The fungal isolates AM.5, AM.8, AM.10, and AM.13 were identified as *P. citrinum*. One isolate, AM.3 was defined as *Paecilomyces* sp. Interestingly, six fungal isolates were identified as *Aspergillus* as follows: three fungal isolates (AM.2, AM.4, and AM.14) were identified as *A. ustus*, two isolates (AM.6 and AM.12) were identified as *A. terreus*, and one fungal isolate (AM.9) was identified as *A. chinensis.* For the first time, *Induratia* (syn. *Muscodor*) (for isolate AM.7) was isolated from deteriorated paper (Figure 7).

As shown, *Aspergillus* spp. and *Penicillium* spp. represented the highest percentages of the obtained strains with equal values at 42.8%, followed by *Induratia* sp. and *Paecilomyces* sp. with percentages of 7.14% for each. Among *Aspergillus* spp., the *A. ustus* was represented by a percentage of 50%, followed by *A. terreus* with percentages of 33.3%, and *A. chinensis* is represented by 16.6%. Interestingly, *P. citrinum* was the most common *Penicillium* spp. isolated from the historical manuscript with a percentage of 66.6% (from the total of *Penicillium* spp.) followed by *P. chrysogenum* with a percentage of 33.3%. Overall, the most common fungal strains isolated from collected deteriorated papers and leather binding belonged to *Aspergillus* spp. and *Penicillium* spp. Similarly, 20 fungal isolates were isolated from old European manuscripts (19th century) and identified using morphological and cultural methods. The authors reported that *Aspergillus* spp. were the most common fungal isolates with a percentage of 45% followed by *Penicillium* spp., *Eurotium* sp., and *Mycelia sterilia* with percentages of 35%, 5%, and 15%, respectively [64]. In a similar study, the most common fungal strains associated with deteriorated paper collected from a historical manuscript dating back to the 17th century were *A. niger* and *A. flavus*, whereas *A. niger*, *A. terreus*, and *A. flavus* were the most common isolates belonging to the leather bookbinding [65]. On the other hand, the fungal strains *A. fumigatus*, *A. tamarii*, *Eurotium chevalieri*, *Cladosporium cladosporioides*, *Fusarium poae*, and *Wallemia sebi* were isolated from leather bookbinding of ancient manuscripts (18th century) [66]. The most common fungal communities isolated from deteriorated manuscripts (79 manuscripts) collected from Astan Quds library, Iran were identified based on their morphological and microscopic characteristics to *Aspergillus* spp. and *Penicillium* spp. [67].

To confirm the traditional identifications, one isolate from each genus was selected and subjected to DNA extraction, amplification, and sequencing of ITS region. Therefore, the fungal strains AM.3, AM.4, AM.5, AM.6, AM.7, AM.9, and AM.11 were selected to be identified by molecular method. The sequence of ITS revealed that the selected fungal strains were similar to *A. ustus*, *A. terreus*, *A. chinensis*, *P. citrinum*, *P. chrysogenum*, *Induratia* sp., and *Paecilomyces* sp. The sequence identity percentages, GenBank accession number, and closest accession number are recorded in Table 1. Hence, the seven selected fungal strains were identified as *A. ustus* AM.4, *P. citrinum* AM.5, *A. terreus* AM.6, *Induratia* sp. AM.7, *A. chinensis* AM.9, *P. chrysogenum* AM.11, and *Paecilomyces* sp. AM.3 (Figure 8). Overall, the molecular identifications were matched with traditional methods by morphological and microscopic examination.

The identification of the many fungi connected with historical manuscripts in libraries, museums, and archives serves a variety of purposes, including the determination of how effective they are in causing biodeteriorations and how harmful they are to workers’ and visitors’ health. As previously reported, several fungal strains that colonize historical papers can cause different human diseases such as dermal infections, allergies, phaeohyphomycosis, and respiratory diseases [68]. Several strains such as *Rhizopus*, *Cladosporium*, *Aspergillus*, *Scopulariopsis*, and *Penicillium* were reported in high percentages in libraries and museums and were considered the main reason for the spread of allergic respiratory infections among workers [69]. Different fungal strains of *Penicillium* and *Aspergillus* are characterized by their ability to cause allergic reactions and respiratory diseases [70]. *A. flavus*, *A. niger*, *A. ustus*, *A. terreus*, *A. udagawae*, *A. alliaceus*, *A. quadrilineatus*, *A. lentulus*, *P. citrinum*, and *P. chrysogenum* can cause a wide variety of diseases such as cutaneous inflammatory, otomycosis, aspergillosis, and endocarditis [53,71]. The presence of fungal strains, especially *Aspergillus* and *Penicillium*, could be attributed to the storage and environmental conditions and the availability of carbon sources (cellulose, starch, xylan; the main components of the historical manuscript) that enhance fungal growth. In the current study, bad storage conditions, temperature (24 °C), high relative humidity (65%), and poor ventilation are considered the main reasons for the adaptation and growth of fungal strains.

### 3.3. Hydrolytic Enzyme Activities

Because of their capacity to grow on paper fibers, produce a variety of hydrolytic enzymes such as amylase, cellulase, pectinase, and xylanase, and/or secrete hydrolytic acids or pigments, fungi are thought to be the primary cause of the biodeterioration of archeological manuscripts [70]. In the paper industry, some supplementations, such as sugars, proteins, starch flours, gelatin, and some artificial additives, are added to decrease the ink spread and enhance the bonding between different fibers [72]. Therefore, fungi can secrete various lytic enzymes that degrade both paper components (cellulosic fibers) and additives, leading to the weakening and deterioration of these materials [73]. Therefore, the efficacy of the isolated fourteen fungal strains to produce hydrolytic enzymes of amylase, cellulase, and pectinase were investigated. Also, their efficacy to producing gelatinase could be related to leather deterioration.

An analysis of variance reported that most fungal strains have the potential to produce cellulase, amylase, gelatinase, and pectinase to various degrees, except for some fungal strains that do not have the efficacy to produce gelatinase and pectinase (Figure 9A–D). As shown, *Induratia* sp. AM.7, *P. citrinum* AM.8, and *A. chinensis* AM.9 (isolated from deteriorated paper samples) could not produce gelatinase enzyme. Whereas the pectinase enzyme was secreted by all strains except *A. ustus* AM.2, *A. ustus* AM.4, *Induratia sp* AM.7, and *A. chinensis* AM.9 (Figure 9C,D).

Data analysis showed that the highest cellulase activity was recorded for *P. citrinum* AM.5 with an inhibition zone of 37.3 ± 4.1 mm followed by *A. ustus* AM.4, *A. terreus* AM.12, and *P. chrysogenum* AM.11 with inhibition zones of 33.3 ± 2.5, 30.7 ± 1.16, and 28 ± 2 mm, respectively (Figure 9A). The cellulase activity among fungal strains AM.1–AM.14 was significant (*p* ≤ 0.001) with inhibition zones ranging between 8.0 ± 0 mm to 37.3 ± 4.1 mm. Recently, *P. chrysogenum* and *A. niger* associated with historical papers dating back to the 17th and 18th centuries have shown the highest activity in cellulase producers [53]. Data recorded by El Bergadi et al. [74] are incompatible with the current study, due to the fact that only nine fungal strains from the 31 strains isolated from a historical manuscript that were collected from Medina of Fez have the efficacy to produce cellulase enzyme. Fungi can penetrate the paper fibers and accumulate various metabolites such as enzymes and acids, leading to physical and chemical alteration [75]. Among 26 fungal strains isolated from the indoor environment of the National Archive of the Republic of Cuba, 88% of these strains exhibited the efficacy to secrete cellulase enzyme leading to alteration in paper crystallinity [76]. Furthermore, the authors reported that approximately 54% and 81% of the total fungal isolates have the potential to deteriorate paper materials through the production of gelatinase and amylase enzymes, respectively. Among 34 *Aspergillus* and *penicillium* strains isolated from indoor cultural heritage conservation sites in Serbia, 16 (47%) and 28 (82%) were characterized by proteolytic and cellulolytic activity, respectively [77].

Data analysis showed that the highest amylase activity was recorded for *P. citrinium* AM.13 and *Induratia* sp. AM.7 with clear zones of 23.3 ± 3.5 and 20.0 ± 2.0 mm, respectively, followed by *P. citrinium* AM.5 and *A. chinensis* AM.9 with clear zones of 19.0 ± 1.0 and 16.7 ± 4.04 mm, respectively (Figure 9B). Interestingly, the amylase activity among fungal strains AM.2–AM.4 was not significant (*p* ≥ 0.001) with clear zones in the range of 10.0 ± 1.0 mm to 13.3 ± 1.5 mm, as well as amylase activity among fungal strains AM.10–AM.12 was also not significant with zones ranging between 6.0 ± 1.0 mm to 7.3 ± 2.5 mm. Our results reveal that the maximum gelatinase enzyme production was recorded for *P. citrinium* AM.13 and *A. ustus* AM.14 with clear zones of 27.0 mm (Figure 9C). *Induratia* sp. AM.7, *P. citrinium* AM.8, and *A. chinensis* AM.9 did not show any gelatinase activity. The resulting analysis showed that the highest pectinase production was recorded for *A. terreus* AM.12 and *A. ustus* AM.14 with clear zones of 33.3 mm (Figure 9D). Interestingly, the pectinase activity amongst fungal strains AM.12–AM.14 was not significant (*p* ≥ 0.001) with inhibition zones ranging between 33.0 ± 1.7 mm to 33.3 ± 2.5 mm. Moreover, *A. ustus* AM.2, *A. ustus* AM.4, *Induratia* sp. AM.7, and *A. chinensis* AM.9 did not exhibit any pectinase activity.

The secretion of the previous hydrolytic enzymes enables the fungal strains to break down large molecules or polymers into small units; for instance, cellulase and amylase enzymes have the efficacy to break down cellulose and starch into glucose monomers. Moreover, a wide range of proteins such as collagen, fibroin, and keratin, used for wool, parchment, silk, and leather, can be degraded to small subunits by the action of protease enzymes such as gelatinase [78]. The ability of isolated fungal strains in the current study to release a variety of hydrolytic enzymes allows us to better understand their effectiveness in the biodeterioration of historical objects (paper and leather bookbinding). Additionally, the storage environment, particularly the relative humidity (65%) and temperature (24 °C) promotes the growth of a variety of fungi and is thought to be an ideal setting for the production of various active metabolites, such as acids and enzymes, as previously reported [70]. In addition to their ability to produce enzymes, isolated fungi are distinguished by their capacity to produce acidic metabolites that promote the degradation of historical manuscripts through acid hydrolysis. According to Borrego et al. [15], the fungal strains *A. niger*, *A. terreus*, *A. ustus*, *A. versicolor*, *P. commune*, *P. chrysogenum*, *P. citrinum*, and *Cladosporium* sp. release acidic metabolites that can lower pH levels by up to four and increase acid hydrolysis.

The contamination of paper by fungal strains can originate from diverse sources such as air, polluting materials, and vectors such as arthropods [79]. Once the environmental conditions are suitable, the fungal spores grow to form the hyphae which germinate and develop to form mature strains. Hence, this mature structure produces various metabolites that enhance biodeterioration. Among deterioration signs are a weakening of the mechanical properties of the paper, difficulty reading, and damage to the information due to pigment secretions. The concentrations of fungal spores in the libraries differ and are influenced by various factors such as season, environmental conditions (RH and temperature), and the number of visitors which may transfer the spore from outdoors to indoors [80].

This research has some limitations. First, just one medium was used for the isolation of the fungal strains, which might have limited the total number of isolates that could be obtained. Second, only in-vitro ionculation of fungal strains over Whatman filter paper were used to explore the metabolic interaction with the materials. Therefore, additional studies should be carried out and are now being investigated to obtain a definitive response to these restrictions.

## 4. Conclusions

In this study, the deterioration aspects of a historical manuscript entitled “Musnad of Imam Muhammad bin Idris al-Shafi’I”, dating back to the 14th century and preserved at the Library of the Arabic Language Academy, Cairo, Egypt, were investigated The most prevalent signs of the leather binding deterioration are darkness, stiffness, loss of flexibility, and missed parts, whereas yellowing, fragility, and weakening are the most prevalent deterioration characteristics of the historical paper. ATR-FTIR, SEM, XRD, and color analyses indicated that the historical samples were exposed to severe deterioration factors compared to the control samples. pH values of historical leather and paper tended to be acidic. Fourteen fungal strains were obtained from the historical manuscript. These strains were identified using traditional and molecular methods as *P. chrysogenium*, *A. ustus*, *P. citrinum*, *A. terreus*, *A. chinensis*, *Induratia* sp., and *Paecilomyces* sp. The isolated fungal strains exhibit high efficacy in producing hydrolytic enzymes including cellulase, amylase, gelatinase, and pectinase which reveal their role in the deterioration and decomposition of the historical manuscript. Based on these findings, it should be recommended to move the damaged manuscripts from their current storage location due to the unpredictable environmental factors. Additionally, staff members need to be trained in how to eliminate degrading catalysts such as dust, mold, rodents, and insects when they appear.

## Figures and Tables

**Figure 1 life-12-01821-f001:**
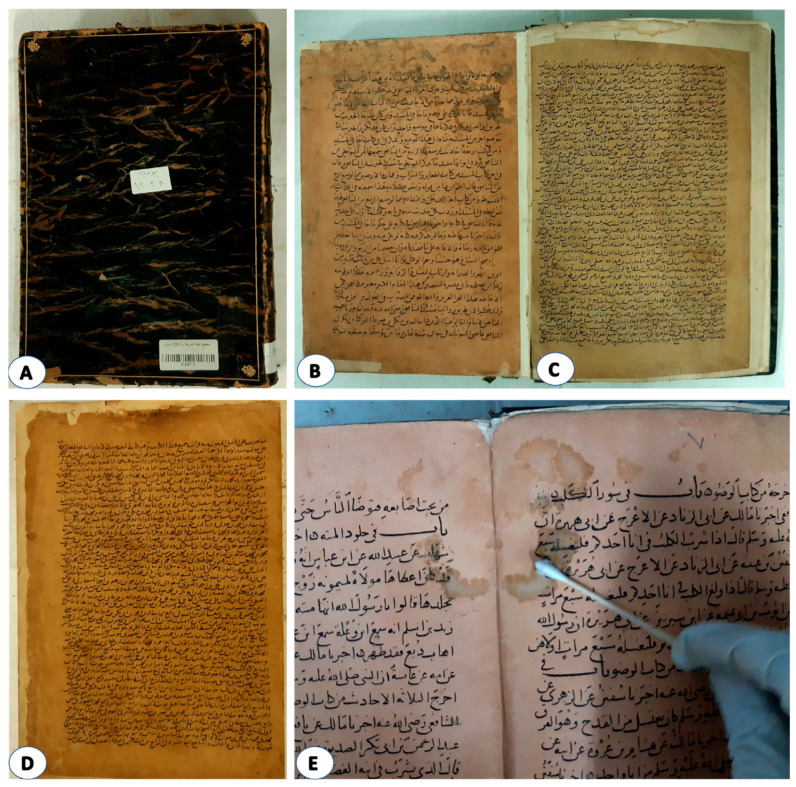
Deterioration aspects of the historical leather binding and paper manuscript: (**A**) Leather bookbinding; (**B**–**E**) Historical paper manuscript.

**Figure 2 life-12-01821-f002:**
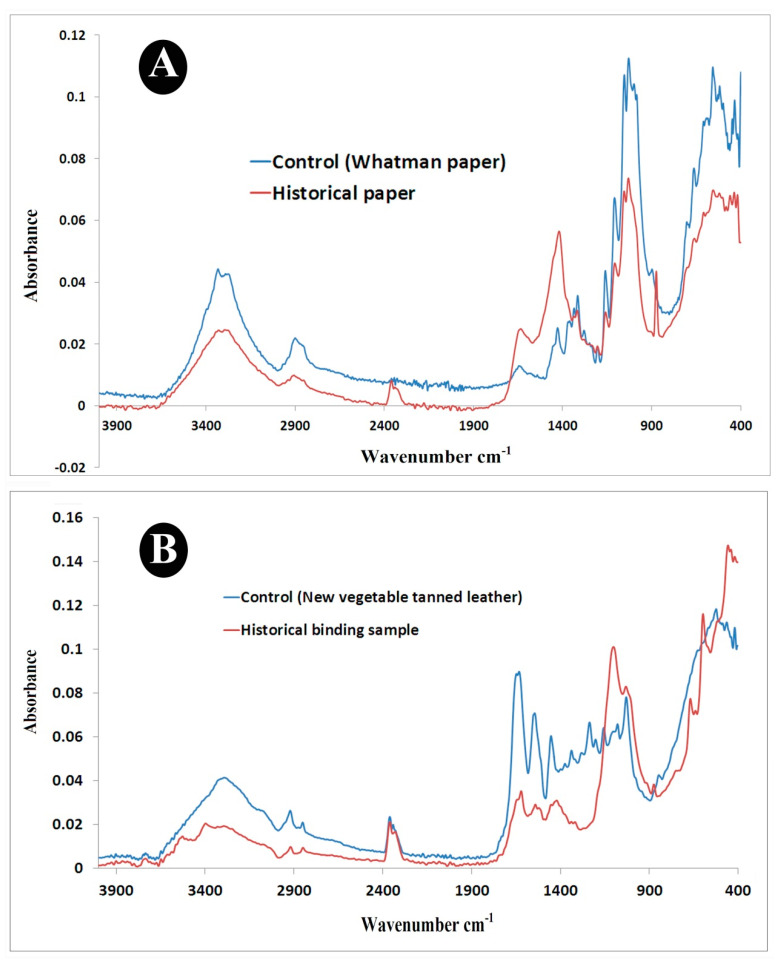
ATR/FTIR analysis: (**A**) Whatman paper and historical paper samples; and (**B**) new, vegetable-tanned leather and historical binding samples.

**Figure 3 life-12-01821-f003:**
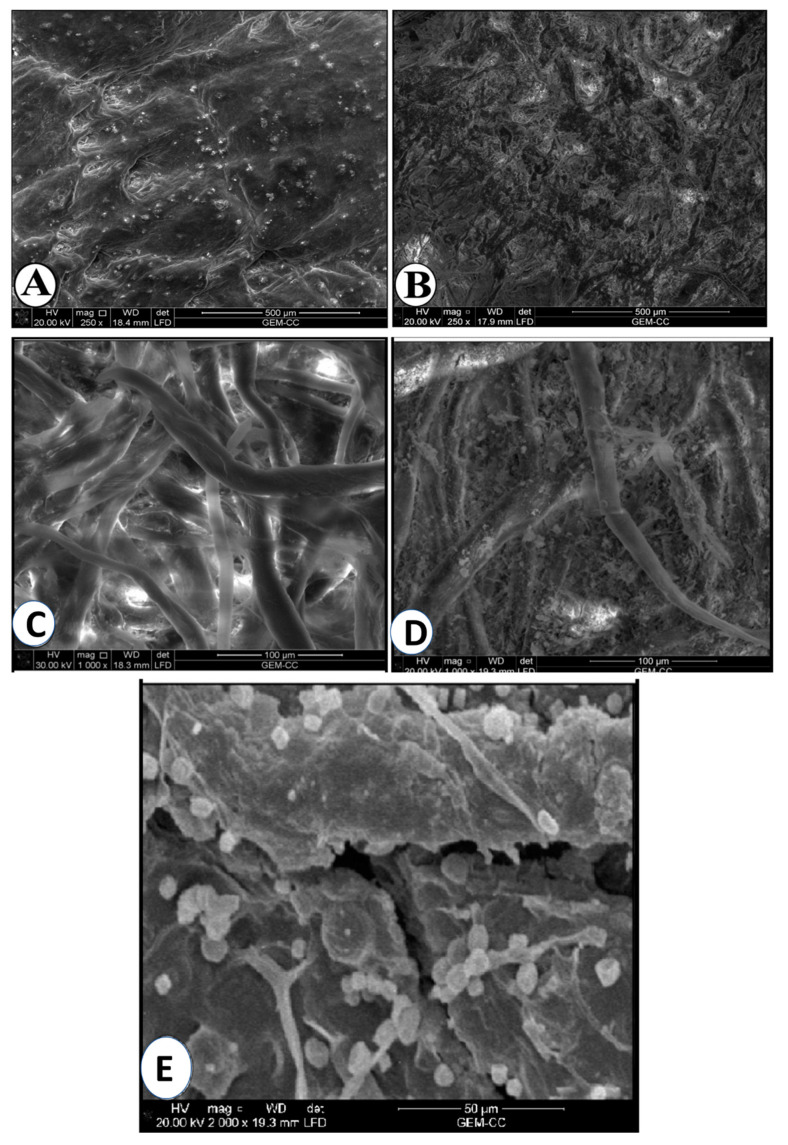
SEM analysis for the surface of historical leather bookbinding and historical paper compared with the control showing the deterioration aspects and growth of fungi. (**A**) SEM image of new goat vegetable-tanned leather sample (control); (**B**) SEM image of historical vegetable tanned leather sample; (**C**) SEM image of new Whatman paper (control); (**D**) SEM image of historical paper; (**E**) SEM image of historical paper showing the growth of fungi.

**Figure 4 life-12-01821-f004:**
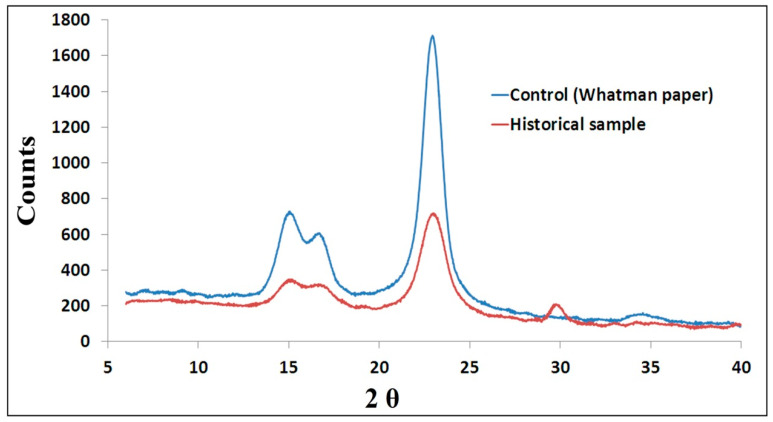
XRD analysis showing the crystallinity index of control (Whatman paper) and historical paper samples.

**Figure 5 life-12-01821-f005:**
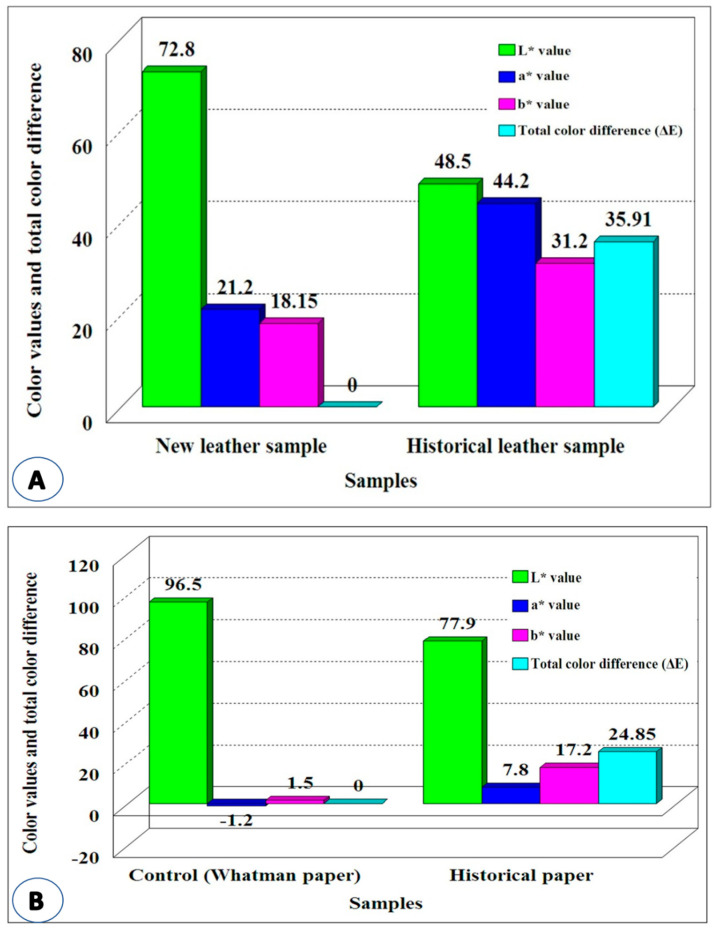
Color change of: (**A**) new (control) and historical vegetable-tanned leather, and (**B**) new (Whatman paper) and historical paper manuscript.

**Figure 6 life-12-01821-f006:**
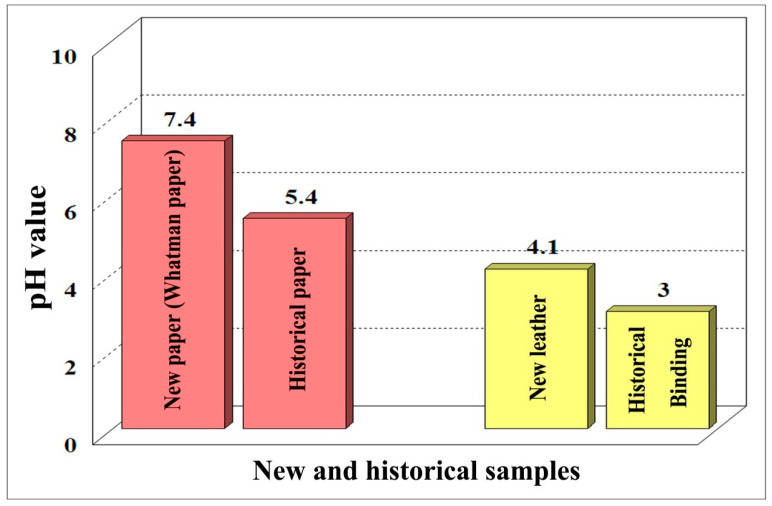
PH values of the new and historical paper and leather samples.

**Figure 7 life-12-01821-f007:**
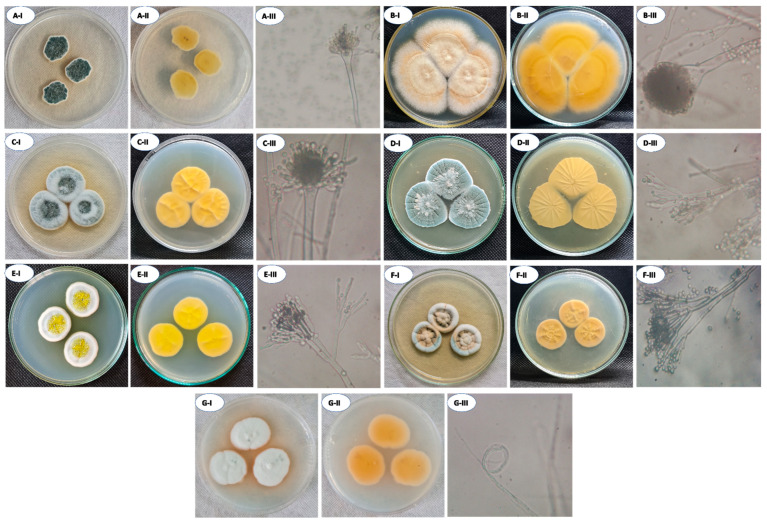
The traditional identification of the isolated fungal strains growing on CYA media based on morphological and microscopic examination. *Aspergillus ustus* (**A**), *Aspergillus terreus* (**B**); *Aspergillus chinensis* (**C**); *Penicillium citrinum* (**D**), *Penicillium chrysogenum* (**E**), *Pacilomyces* sp. (**F**), *Induratia* sp. (**G**) CYA colony observation (**I**), CYA colony reverse (**II**), and conidiophore (**III**) (magnification power = 800).

**Figure 8 life-12-01821-f008:**
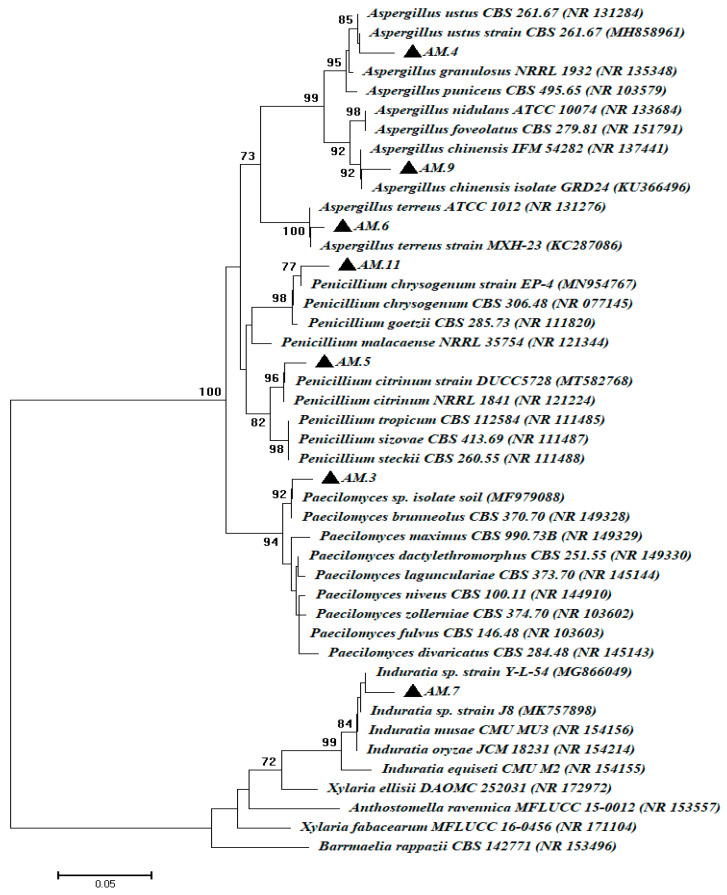
Phylogenetic tree of seven selected fungal strains identified by ITS sequences compared with reference sequences from NCBI. The phylogenetic tree was achieved by neighbor-joining method and bootstrap value of 1000 replicates.

**Figure 9 life-12-01821-f009:**
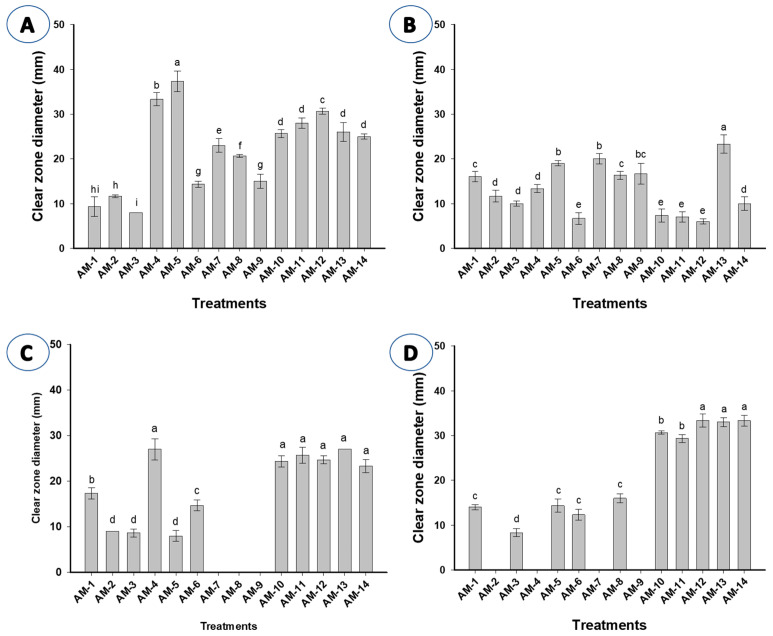
The efficacy of fungal strains isolated from a historical manuscript (paper and leather bookbinding) to secrete various enzymes. (**A**) cellulase activity, (**B**) amylase activity, (**C**) gelatinase activity, and (**D**) pectinase activity.

**Table 1 life-12-01821-t001:** The identification of fungal strains using ITS analysis, sequence identity percentages, and GenBank accession number.

Fungal Code	GenBank Accession Number	Homolog Sequences (Sequence Identity%)	Closest Accession Number
AM.3	ON527926	*Paecilomyces* sp. (98.96%)	MF979088
AM.4	ON527927	*Aspergillus ustus* (98.82%)	NR131284
AM.5	ON527930	*Penicillium citrinum* (98.53%)	NR121224
AM.6	ON527928	*Aspergillus terreus* (98.21%)	NR131276
AM.7	ON527932	*Induratia* sp. (98.76%)	NR154156
AM.9	ON527929	*Aspergillus chinensis* (99.01%)	NR137441
AM.11	ON527931	*Penicillium chrysogenum* (98.24%)	NR077145

## Data Availability

The data presented in this study are available on request from the corresponding author.

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
