# Peer review of "Fungal Biodeterioration of a Historical Manuscript Dating Back to the 14th Century: An Insight into Various Fungal Strains and Their Enzymatic Activities"

_life, 2022, doi:10.3390/life12111821_

Round 1

Reviewer 1 Report

I have read the manuscript entitled ''Fungal Biodegradation of Historical Manuscript Dated Back to 2 the 14th Century, An Insight into Various Fungal Strains and 3 Their Enzymatic Activities'' (Manuscript ID: life-2002391). This is an interesting study which examines biodeterioration of cultural heritage by fungi. The authors have employed various techniques, reported the fungal diversity and demonstrated biodegradation potential of fungal isolates. However, the manuscript is poorly written and requires major changes before publishing.

General comments

Biodegradation is defined as a process in which microorganisms are used to modify material with a positive or useful outcome. On the other hand, biodeterioration refers to the undesirable or negative changes to the material. Therefore the authors should write "biodeterioration" when referring to the negative impact of living organisms on the manuscripts. Likewise, the term "biodegradation" can be used when addressing in vitro experiments such as the production of enzymes by fungal isolates.

The manuscript requires thorough English language editing.

Specific comments

Title

I would replace the term "biodegradation" with the term "biodegradation".

Simple Summary

Page 1, line 18 – replace "objectives" with "objects".

Page 1, line 19 – replace "degradation" with "deterioration".

Page 1, line 20 – replace "and hence" with "which leads to the".

Page 1, line 20 – replace "to secretion" with "to secrete".

Page 1, line 27 - replace "to producing" with "to produce".

Abstract

Page 1, lines 34-35 – replace "and explored their activity in deterioration" with "and their deterioration activity was assessed".

Introduction

Poor quality of English language. Inadequate style of writing (you should rephrase sentences in the lines 63-66).

Page 2, lines 69-72 - You refer to the Sterlinger as HE, but it is in fact a female researcher. You should avoid addressing the claims of other authors in this way.

Page 3, lines 108-109 – Please delete the fragment:" compared to the controls (Whatman paper No. 1 for paper and vegetable tanned goat skins for leather bookbinding)".

Page 3, line 114 - replace "biodegradation" with "biodeterioration".

Materials and methods

Page 5, lines 232-233 – state which statistical method was used for phylogenetic analyses. Which software was used to perform these analyses?

Page 6, line 248 - replace "achieved" with "performed".

Results and discussion

Poor quality of English.

Page 16, line 547 - replace "inhabitant" with "inhabit"

Page 16, line 562 – If you refer to the fungal genus Muscodor, its current name is Induratia, and therefore you should replace it throughout the manuscript. You can write Induratia (syn. Muscodor) when referring to this taxon for the first time. Note that you are misspelling this genus name throughout the manuscript (Muscodora, see lines 562 and 565-566).

Page 17, line 584 - replace "fumigates" with "fumigatus"

Page 18, lines 594-597 – please reformat the caption if possible, such as: Aspergillus ustus - colony obverse (A) and reverse (B) and conidiophore (C and D) etc. All light microscopy micrographs are of the bad quality, blurry, with the presence of blackish artifacts and all lack scale bars – you mentioned magnification but it is obvious that magnifications are not the same (for example A3 and A4). Consider using only one micrograph of a good quality instead of two.

Page 19, lines 598-615 – consider putting percentages of homologies and gene bank accession numbers in a table since it is hard to follow them thorough the text. Consider revising Table 1 for this purpose. You do not need a separate table to show the results of traditional identification but rather the one which will show the final identification results obtained by both traditional and molecular approach.

Page 19, lines 598-615 - Identification of Aspergillus, Penicillium and Talaromyces species by using only ITS region as a molecular marker is not sufficient. The species of these genera can only be correctly identified if using appropriate alternative genes such as beta tubulin gene (BenA). I advise to perform additional sequencing for the correct identification of these species.

Page 19, lines 615-616 – Consider deleting this sentence.

Page 19, lines 625-626 – Put the names of the genera in italic.

Page 19, line 621 – Replace "can be causes" with "can cause".

Page 19, lines 633-634 – Why is 24 ºC considered a low temperature? Could you elaborate what are recommended environmental parameters (T and RH) for the storage of the manuscripts in archives and libraries and provide a suitable reference.

Page 20, lines 637-639 – You should write which statistical method you used for the construction of phylogenetic three, was it Maximum Likelihood, Neighbor Joining or something else?

Page 21, line 655 - replace "which isolated" with "which were isolated".

Page 21, lines 666-670 – Please discuss cellulolytic potential of fungal isolates this more thouroghly and compare it to the results of other studies. See the references:

Anaya, M., Borrego, S. F., Gámez, E., Castro, M., Molina, A., & Valdés, O. (2016). Viable fungi in the air of indoor environments of the National Archive of the Republic of Cuba. Aerobiologia32(3), 513-527.

Savković, Ž., Stupar, M., Unković, N., Ivanović, Ž., Blagojević, J., Vukojević, J., & Ljaljević Grbić, M. (2019). In vitro biodegradation potential of airborne Aspergilli and Penicillia. The Science of Nature106(3), 1-10.

Given the abovementioned comments, I suggest a major revision before acceptance to the Life.

Author Response

Dear reviewer, many thanks for your valuable comments. We answered all comments point by point as shown in the uploaded author response.

Reviewer 2 Report

The paper suffers from innumerable weak points and scientific inaccuracies.

From an experimental point of view, the greatest criticism concerns the choice of controls, which appear inadequate to judge the state of degradation of such an ancient manuscript. It is in fact insignificant to compare the chemical-physical characteristics of such ancient materials with contemporary materials, without these even having been artificially aged using standardised methods. The best control should be a coeval material in an optimal state of preservation. In the absence of this, controls consisting of contemporary material should at least be artificially aged.

Analytical methods are acceptable, although not particularly innovative, nor fully informative. The main problem concerns the lack of indication of the quantity of samples obtained per analysis from the artefact and the lack of description of the chosen sampling points.

More specifically, with regard to the microbiological analysis method (which cannot be defined as an independent culturable technique, as erroneously indicated in the abstract), the main criticism concerns the choice of subjecting the samples to an enrichment culture. Due to the type of micro-organisms being analysed (biodeteriogenic environmental moulds) and the type of material from which they are taken, the enrichment method is at high risk of false positives.

The method of pH analysis on paper is not well explained. The one performed on leather, on the other hand, is destructive: for such ancient and precious objects, destructive sampling is only acceptable for erratic fragments that cannot be relocated. In this case it is not clear how and where the sampling was done.

The results do not seem sufficient to support the countless speculations in the paper on the mechanisms of biological degradation of library materials.

The few organisms obtained from the material do not demonstrate their role in the degradation of the substrates, which may also have occurred previously. On the other hand, no tests were performed to verify the state of metabolic activity of these organisms directly on the materials.

Author Response

(The authors gave the same response as above.)

Reviewer 3 Report

Manuscript ID life-2002391 deals with study of deterioration process of 14th century historical manuscript, in particular with role of fungi isolated from the manuscript and characterization of their enzymatic potential. The study is well designed and of interest to the broad spectrum of readers, especially those involved in the science of biodeterioration. However, in present state there are some flows that need to addressed before the manuscript can be accepted for publication.   

- English leaves a lot to be desired for. I suggest authors to seek native speaker to proofread the manuscript and correct all of the mistakes. There are too many to list.

- In present state, manuscript has too many technical errors, and was probably not quality checked before submission. I recommend authors to carefully read the manuscript and follow guidelines of the journal before resubmission.

- I suggest to replace term "biodegradation/degradation" with "biodeterioration/deterioration" whenever addressing symptoms present on the manuscript. Term "degradation" is acceptable for use only when referring to the decomposition of compounds objects are made of, irrespectively of the source of degradation.

- There is no need to constantly at every opportunity introduce the same abbreviation again and again. Introduce it at the first opportunity an then use only abbreviated form (e.g. SEM, XRD...).

Lines 41-42: I would not list P. citrinum after Paecilomyces genus since uninformed reader might get an impression that P. citrinum refers to species of  Paecilomyces genus when in fact it belongs to Penicillium genus. Correct in all places in the manuscript.

Lines 69-73: Sterflinger is a female. Her name is Katja.

- sp. or spp. is never written in italic. Correct in all places.

- There is no need to constantly repeat what the control samples are. Mention in once in the Material and Methods section at after that it is implied that you refer to it.

Lines 141-142: How were these parameters measured?

Line 149: Digital camera is not used for visual assessment. It is used for photo documentation that is analyzed and assessed. 

- Why was only one general nutrient medium used for isolation of fungi? And why did you not use specific media designed for isolation of fungi from paper? You would have isolated more specific fungi that way.

- How was species of genus Muscador identified? What key was used?

Lines 258-260: darkness? missed parts? I urge authors to consult an expert and use appropriate terminology used in the science of biodeterioration when describing symptoms.

- Figure 2A is too far away from the paragraph that it is first mentioned in.

- In Results section, do not give chapter names per used methodology but per obtained results.

Lines 405-406: Mycelium is composed of hyphae.

- Figure 3 legend: SEM is mentioned six times. There is no need for that.

Lines 519-524: Many of the mentioned genera are not genera of cellulolytic fungi but genera where some members possess cellulolytic enzymes. Refer to them as such and distinguish them from true cellulolytic fungi such as Chaetomium spp.

Line 547: I would not use the term "inhabit" as it implies colonization and active growth. Where there any visible symptoms of fungal growth? If there were none, then isolated fungi were merely contaminants of paper present on the surface in form of propagules.

Figure 7 legend: Please use correct mycological terminology when describing obverse and reverse side of colonies.

Line 598: On what basis was selection made? What were the criteria?

Lines 598-616: There is no need for accession numbers, they only burden the text. Remove them.

- Names of fungal genera are always written in italic. Correct where it is not written that way.

Line 620: Contaminate not colonize.

Line 707-709: This statement is incorrect. Spores germinate into hyphae that grow to form mycelium. Strains are something completely different.

Author Response

(The authors gave the same response as above.)

Round 2

Reviewer 1 Report

The manuscript is now much improved. I suggest accepting.

Author Response

Dear reviewer, Many thanks for your agreement and your approval

Reviewer 2 Report

Author response to Reviewer (2) comments

(life-2002391)

Thank you for reviewing our manuscript and giving us helpful comments. We made corrections and we hope they meet with your approval. We revised the paper according to your specific comments. Details explanations for the comments are shown below.

Reviewer comment: The paper suffers from innumerable weak points and scientific inaccuracies.

Author response: Dear reviewer, thank you for your valuable comments. We revised the manuscript according to all reviewer comments and we hope the revised version meets your approval.

Reviewer comment: From an experimental point of view, the greatest criticism concerns the choice of controls, which appear inadequate to judge the state of degradation of such an ancient manuscript. It is in fact insignificant to compare the chemical-physical characteristics of such ancient materials with contemporary materials, without these even having been artificially aged using standardised methods. The best control should be a coeval material in an optimal state of preservation. In the absence of this, controls consisting of contemporary material should at least be artificially aged.

ü Author response: Thank you very much for your comment, but please let us to say the following points concerning this comment:

v Modern standard samples of Whatman paper No. 1 and vegetable tanned leather were selected and compared to historical samples as an analytical study (but not as an experimental study) in order to know the extent of the destruction of the historical manuscript. v On the other hand, the objective of artificial accelerated ageing is always applied on modern standard samples to prepare aged samples similar to historical or archaeological samples to be used in experimental studies to evaluate some conservation treatment methods and materials which can be applied to the historical manuscript. It should be mentioned that according to international conventions and legislation recommend not to use of archaeological or historical materials in experimental studies to evaluate conservation treatment materials, but experimentation is carried out on modern artificially aged samples, and the materials that gave high efficiency in the conservation treatment and concluded from the evaluation process is applied directly to the archaeological or historical manuscript. v Accordingly, the aim of this study is to know the damage found in the manuscript so that treatment processes that were evaluated on artificially aged modern samples can be applied to the historical manuscript (this will be the next publication). 

Reviewer response

See point by point comment

Reviewer comment: Analytical methods are acceptable, although not particularly innovative, nor fully informative. The main problem concerns the lack of indication of the quantity of samples obtained per analysis from the artefact and the lack of description of the chosen sampling points.

Author response: Thank you very much for this comment. But it can be said that although the methods of analyses are not innovative, but they are highly efficient in detecting deterioration and its mechanism of the manuscript studied. Most of the analytical techniques used are considered non-destructive such as ATR/FTIR, pH value of paper, color change, and fungal investigation. Some other methods are considered micro-destructive such as measuring the pH value of leather, SEM, and XRD (minutes of samples are needed, and the samples were taken from paper or leather fibers that fallen next to the manuscript but will not be returned back to it). It can also be said that the analyses used in the current study are recommended by scientists and researchers in the field of conservation of paper and restoration studies either in the experimental studies for the evaluation of conservation methods and materials or in the analytical studies on historical or archaeological manuscripts such as current study. It is known either in the current study or in other studies that the samples always were taken from damaged places with severe aspects of deterioration.

Reviewer response

Ok, so I suggest you to insert the sentence “the samples always were taken from damaged places with severe aspects of deterioration” in the text too. Furthermore it would even better to associate any sample point with any analytical method and quantify them. The reader can get an idea of the type of degradation only from the very few images you present in figure 1, where I can observe few different types of degradation such as embrittlement, tideline, oxidation…. Where did you exactly perform your analysis?

You can for example insert a table like this

analysis

sample

N° of sampling point

Description of the damage

substrate

Detail of the damage

ATR-FTIR

1

3

tideline, discoloration

paper

ATR-FTIR

2

3

Fungal attack

paper

SEM

3

2

embrittlement

leather

COLOR MEASUREMENT

4

3

oxidation

paper

Ecc.

This is just an example.

It should be clear the association between damages and results obtain from any analytical investigation. Otherwise your output turns out to be too much generic and obscure.

By the way, I can’t notice any sign of fungal attack on leather binding, at least on the pictures reported in your paper

XRD, SEM were performed on minutes of materials taken from the book. You have to specify in the text

Reviewer comment: More specifically, with regard to the microbiological analysis method (which cannot be defined as an independent culturable technique, as erroneously indicated in the abstract), the main criticism concerns the choice of subjecting the samples to an enrichment culture. Due to the type of micro-organisms being analysed (biodeteriogenic environmental moulds) and the type of material from which they are taken, the enrichment method is at high risk of false positives.

Author response: We apologize for this mistake, we used dependent culturable technique, not independent methods. So, this mistake was corrected in the revised version. Regarding enrichment broth media, we used to enhance the growth of fungal strains that colonized historical sample before inoculated into solid media. This method is common for isolation of such fungi (such as Aspergillus, Penicillium, Pacilomyces) form historical manuscripts as mentioned in literature. Please see the following literature that used the same sequence for isolation of fungi colonized historical paper: (El Bergadi, F., Laachari, F., Elabed, S. et al. Cellulolytic potential and filter paper activity of fungi isolated from ancients manuscripts from the Medina of Fez. Ann Microbiol 64, 815–822 (2014). https://doi.org/10.1007/s13213-013-0718-6),

Reviewer response:

I take note of the reference cited, but most of the papers you can find in this field doesn’t use culture enrichment. Maybe you can incubate for few hours your swab in physiological solution just for rehydration and spore reactivation (see for example Pinzari, F., Montanari, M., Michaelsen, A., and Pinar, G. (2010). Analytical protocols for the assessment of biological damage in historical documents. Coalition Newsletters 19: 6-12 (ISSN 1579-8410 5, www.rtphc.csic.es/boletin.htm)), but if you add sugar as in your Czapeck broth the risk of false positive is too high.  Many biodegraders, including those isolated in the present study, are common environmental organism growing on many organic substrates and also dust. They are also common contaminant of axenic culture. Enrichment may enhance and multiply occasional spores that are on the surface by chance without any biodeterioration role. Furthermore you don’t make any attempt to find correspondence among spores visualized by SEM on the surface and strain obtained in culture.

Anyway, I admit the presence of different opinions and references, so I accept your response.

 Reviewer comment: The method of pH analysis on paper is not well explained. The one performed on leather, on the other hand, is destructive: for such ancient and precious objects, destructive sampling is only acceptable for erratic fragments that cannot be relocated. In this case it is not clear how and where the sampling was done.

ü Author response: The method of pH measurement on paper was explained according to Abdel-Maksoud [24] with little modification. For measurement of pH value of new and historical leather samples the following text was added to the subsection 2.2.6.1. Measurement of pH value of the Leather samples:

" Mechanically, a sample (0.025 g) of the leather bookbinding was removed from as close to the damaged location as possible in the form of loose fibers. To allow the ions to transfer into the solution, the leather samples were soaked in deionized water for about 6 hours".

It was clear from the method explanation that this method is considered micro-destructive testing, which is accepted in the conservation field because minutes of the sample is needed, especially since we did not cut the sample from the bookbinding, but it was taken from the loose fibers which cannot be returned to the bookbinding.

Reviewer response

See in point by point comment

Reviewer comment: The results do not seem sufficient to support the countless speculations in the paper on the mechanisms of biological degradation of library materials.

Author response: Dear reviewer, among biodeterioration mechanisms is enzymes which used to breakdown of polymer to monomer. In the current study we used deteriorated historical manuscript and measurement the destructive percent by various methods. Also, we used deteriorated parts as a source for isolation of various fungal strains and investigated the activity of these fungi to secretion various hydrolytic enzymes (cellulase, amylase, protease, and gelatinase), and we returned this hydrolytic activity to be a mechanism for biological degradation. The limitation of this paper is represented by the growth of isolated fungi on paper in-vitro and investigate the activity to deteriorate this paper. Actually, we studied this point and will be published based on this paper.

Reviewer response

Yes as you say in vitro analysis on the isolated strains are not completely informative to speculate on in vivo metabolism on the manuscript. You should specify better this point in your paper, also in your last version.

See also in point by point comment

Reviewer comment: The few organisms obtained from the material do not demonstrate their role in the degradation of the substrates, which may also have occurred previously. On the other hand, no tests were performed to verify the state of metabolic activity of these organisms directly on the materials.

Author response: Thank you for your comment. In the current study, we obtained fourteen fungal strains from small, deteriorated part collected from historical manuscripts by dependent culturable method and this number is acceptable according to literature. You know the difficulties to collect such samples. As mentioned in the above comment, the limitation of the current study is verifying the metabolic activity of these organisms on the material in-vitro. We clarify this meaning in the manuscript as follows: “This research has some limitations. First, just one medium was used for the isolation of the fungal strains, which might have limited the total number of isolates that could be obtained. Second, examine how isolated fungal cultures metabolically interact with the materials. Third, utilizing eco-friendly biologically active substances to limit fungal growth and preserve or improve the standard of historical manuscripts. Therefore, additional studies should be carried out and are now being investigated to obtain a definitive response to these restrictions.

Reviewer response

This last sentence that you add in the paper is not very clear, apart the first point (only one medium, yes it’s not so much). See also comments below

Finally, We hope the our response met the reviewer endorsement.

Point by point comment

44: maybe you can avoid to specify “dependent-culturable technique”

111: Gelatinase is a protease. Sometime you mention protease and sometime pectinase, that are very different enzymes with different activity on organic substrates. Protease, such as gelatinase, must be important for fungi isolated from leather. Pectinase could be important only for paper, even if it doesn’t seem so important for such old materials.

124. The purpose is to assess the degradation brought about by fungi. But we have to consider that organic material over such a long time undergoes many other natural degradation processes due to intrinsic and extrinsic factors. Therefore to distinguish and quantify biological damage from natural decay due to ageing the comparison I would make is between the naturally aged sample (the manuscript) damaged by microorganisms and artificially aged samples

155. Specify how the samples were obtained, from where (see my example of table 1) and how they were prepared for SEM observation. Have the samples been metallized? Quanta 3D 200i is also “environmental”. Did you observe in high or low vacuum?

166. again, it’s not clear where did you perform the analysis. From the few pictures I hardly distinguish fungal attack, tideline, oxidation… The reader should know what forms of degradation the values obtained from the analysis correspond to.

190. sorry I don’t find any  enemy 1030 189 pH/mV device in internet. Are you sure about the model?

213: ITS genes doesn’t exist. In fact you are amplifying sequences within rRNA genes. ITS are short region located between different rRNA subunit genes.

216: remove “genes”. You are actually amplifying only a small portion of the large subunit rRNA gene.

235: so here you are testing for pectinase and not for protease?

278: The purpose of your work is not clear. Are you analyzing the general state of degradation of the manuscript or only the degradation caused by microorganisms? In the abstract you claim: “This study aims to assess the characteristics of deterioration caused by fungi that are associated with a historical manuscript dated back to the 14th century that was deposited in the Library of the Arabic Language Academy, Cairo, Egypt, and has undergone fungal deterioration”.

However, in this long chapter (3.1.1.) you talk about many other intrinsic and environmental degradation factors and different types of nonbiological damage, confusing the reader about the ultimate purpose of the paper and the results obtained.

As mentioned above, it is essential to associate the results of the analysis with the forms of degradation found, otherwise it is impossible to speculate on the role of microorganisms in causing certain types of damage on paper and leather.

381: from this sentence it doesn’t seem that fungi are involved in lather degradation as you mention in the abstract

382: it’s not necessary to specify this in the result because it’s not a result. Whatman grade 1 it’s already certified as pure cotton paper

440: natural heat aging? This sentence is obscure. Why “heat”?

491. you’re discussing leather… you should split the paragraphs

495: organic, repetition

581: what does it mean X=800. If you mean magnification of microscopic images you should specify better. In any case it would be better having the dimensional bar on the pictures.

587: again ITS genes doesn’t exist

628: the reference 71 deal with fungal infection in human patient. There is not correspondence with the sentence

632: gelatinase deals with the degradation of leather so you should not mention it when you are discussing on paper degradation.

633: are you sure that pectine is a significant component in archeological manuscript? Have you a reference about this? Your reference 70 doesn’t mention pectine at all in the text. Did you detect pectine in the manuscript?

647: why incompatible? Explain better

680: pectinase doesn’t degrade proteins! It’s seem there is confusion between pectinase and protease

703: this second sentence is not clear

704: not clear. Treatment evaluation has nothing to do with the limitations of your work

732: frequent treatments such those you mention are never suggested if your prevention measures are done correctly  

Author Response

Dear reviewer, Many thanks for your valuable comments. We answered all comments point-by-point as shown in the uploaded author response. We hope the revision meets with your approval. 

Reviewer 3 Report

Dear editor, the authors have made significant changes to the original manuscript ID life-2002391, and it is now suitable for publication in your respected journal.

Author Response

(The authors gave the same response as above.)
